# Enhancing Predictive Tools for Skeletal Growth and Craniofacial Morphology in Syndromic Craniosynostosis: A Focus on Cranial Base Variables

**DOI:** 10.3390/diagnostics15131640

**Published:** 2025-06-27

**Authors:** Lantian Zheng, Norli Anida Abdullah, Norlisah Mohd Ramli, Nur Anisah Mohamed, Mohamad Norikmal Fazli Hisam, Firdaus Hariri

**Affiliations:** 1Department of Oral and Maxillofacial Clinical Sciences, Faculty of Dentistry, Universiti Malaya, Kuala Lumpur 50603, Malaysia; 2Mathematics Division, Centre for Foundation Studies in Science, Universiti Malaya, Kuala Lumpur 50603, Malaysia; 3Center for Data Analytics Consultancy and Services, Faculty of Science, Universiti Malaya, Kuala Lumpur 50603, Malaysia; 4Department of Biomedical Imaging, Faculty of Medicine, Universiti Malaya, Kuala Lumpur 50603, Malaysia; 5Institute of Mathematical Sciences, Faculty of Science, Universiti Malaya, Kuala Lumpur 50603, Malaysia

**Keywords:** syndromic craniosynostosis, midface, cranial base, predictive model

## Abstract

**Background/Objectives**: Patients with syndromic craniosynostosis (SC) pose a significant challenge for post-operational outcomes due to the variability in craniofacial deformities and gain-of-function characteristics. This study aims to develop validated predictive tools using stable cranial base variables to predict changes in the midfacial region and explore the craniofacial morphology among patients with SC. **Methods**: This study involved 17 SC patients under 12 years old, 17 age-matched controls for morphological analysis, and 21 normal children for developing craniofacial predictive models. A stable cranial base and changeable midfacial variables were analyzed using the Mann–Whitney U test. Pearson correlation identified linear relationships between the midface and cranial base variables. Multicollinearity was checked before fitting the data with multiple linear regression for growth prediction. Model adequacy was confirmed and the 3-fold cross-validation ensured results reliability. **Results**: Patients with SC exhibited a shortened cranial base, particularly in the middle cranial fossa (S-SO), and a sharper N-S-SO and N-SO-BA angle, indicating a downward rotation and kyphosis. The midface length (ANS-PNS) and zygomatic length (ZMs-ZTi) were significantly reduced, while the midface width (ZFL-ZFR) was increased. Regression models for the midface length, width, and zygomatic length were given as follows: ANS-PNS = 23.976 + 0.139 S-N + 0.545 SO-BA − 0.120 N-S-BA + 0.078 S-SO-BA + 0.051 age (R^2^ = 0.978, RMSE = 1.058); ZFL-ZFR = −15.618 + 0.666 S-N + 0.241 N-S-BA + 0.155 S-SO-BA + 0.121 age (R^2^ = 0.903, RMSE = 3.158); and ZMs-ZTi = −14.403 + 0.765 SO-BA + 0.266 N-S-BA + 0.111 age (R^2^ = 0.878, RMSE = 3.720), respectively. **Conclusions**: The proposed models have potential applications for midfacial growth estimation in children with SC.

## 1. Introduction

Craniosynostosis, a congenital disorder characterized by the premature fusion of one or more cranial sutures, affects approximately one in every 2500 live births. Among these, syndromic craniosynostosis (SC), which constitutes 27% of cases [1], presents a more complex challenge due to the association with various genetic syndromes such as FGFR, TWIST, and EFNB1 mutations [2]. Common types of SC, such as Apert, Crouzon, and Pfeiffer syndromes, are primarily caused by mutations in FGFR gene mutations. These syndromes are typically marked by an abnormal head shape and are associated with midface deformities, resulting in both aesthetic and functional impairments. Patients with these conditions often require multiple surgical interventions to address their craniofacial abnormalities. However, the complexity and variability of craniofacial deformities in syndromic cases increase significantly because of the diversity of genetic mutations and deteriorate with age due to gain-of-function characteristics and resynsostosis of craniofacial sutures [3]. All these factors complicate the achievement of favorable surgical outcomes and highlight the necessity for individualized treatment strategies.

In SC, the anatomical and developmental interrelationships between the cranial base and the midface are interlinked. The cranial base serves not just as a structural support but also significantly influences overall facial aesthetics and functional outcomes. Reitsma and colleagues observed that altered growth of the cranial base in patients with SC plays a crucial role in resultant midface hypoplasia [4]. Lu et al. further emphasized that premature fusion of the cranial base sutures initiates a chain reaction, leading to severe midface retrusion [5]. These findings suggest that cranial base variables could serve as effective predictors of midfacial developmental outcomes.

Recently, Hariri et al. provided a foundation for predicting maxillary growth using the whole cranial base in a limited sample of the SC population. However, the study had limited age-matching between the SC patient group and the control group [6]. In the current study, this limitation was addressed by ensuring age-matching and including age as a predictor. The aim of this study is to propose predictive tools using stable cranial base variables, with the addition of age, to predict changes in different parts of midfacial outcomes and explore the craniofacial morphology in patients with syndromic craniosynostosis, which could benefit personalized treatment strategies, potentially reducing the need for multiple corrective surgeries, thereby obtaining the achievement of favorable surgical outcomes.

## 2. Materials and Methods

This study was approved by the Medical Research Ethics Committee, University Malaya Medical Center (2022916-11546). Between November 2015 and December 2023, a total of 17 CT scans of non-operated patients with syndromic craniosynostosis (SC) under 12 years old, at a craniofacial unit of a tertiary healthcare center, were selected for this study. For the control group, CT data from 17 age-matched non-syndromic children were collected for the craniofacial morphology analysis. In addition, for the development of predictive growth models, we analyzed CT scans from 21 non-syndromic children under 12 years old that included the same 17 age-matched non-syndromic controls previously used in the morphological comparison. The exclusion criteria of the SC subjects were as follows: (1) non-syndromic and isolated craniosynostosis; (2) incomplete skull and midfacial region images; (3) age over 12 years old; and (4) undergone craniofacial surgeries. The exclusion criteria of the control subjects were as follows: (1) with existing or previous skull deformity; (2) any midface hypoplasia associated with other diseases; (3) incomplete skull and midfacial region images; and (4) age over 12 years old.

The morphometric analysis included midface and cranial base dimensions. There were 4 points in the cranial base area (N, S, SO, BA) that were commonly used in previous papers [7,8,9,10] and 7 points (ANS, PNS, ZMs(L/R), ZTi, ZF(L/R)) that constructed the length, width, and height in different midfacial regions were selected. The non-statistical difference in cranial base variables and statistical difference in midfacial measurements were selected for skeletal growth predictive models. The definition of all the landmarks is summarized in Table 1.

The CT scans, in Digital Imaging and Communications in Medicine (DICOM) format and with a slice thickness of ≤1.0 mm, were processed using the Materialise Mimics Medical software version 21.0 (Materialise, Leuven, Belgium) to reconstruct the 3D images automatically. The approximate location of the landmarks was selected on the reconstructed 3D image, and the final position is adjusted in multiplanar views showing the axial, frontal, and sagittal very carefully. To ensure the reliability of landmark measurements, two independent observers performed the measurements twice on different days (one is a maxillofacial surgeon and the other is a biomedical engineer). Pearson correlation coefficients of both inter-observer and intra-observer were more than 0.80.

The Statistical Package for the Social Sciences (IBM SPSS Statistics, version 25) was utilized to perform the non-parametric Mann–Whitney U test to compare the morphological characteristics and identify the stable cranial base and changeable midfacial variables between the two groups. The statistical significance threshold was set at α = 0.05. R Studio (R 4.3.2 GUI 1.80 Big Sur Intel build (8281)) was utilized for the comprehensive development of the craniofacial predictive models. This involved several key steps: (1) The results from the morphological analysis that identify the stable cranial base and changeable midfacial variables will be used. A correlation analysis was conducted between the midface of ANS-PNS, ZFL-ZFR, and ZMs-ZTi (response variables) and the cranial base of S-N, SO-BA, N-S-BA, and S-SO-BA and age (predictors variables) to ensure any indication of linear relationships. (2) The multicollinearity among the predictors was then checked using the Variance Inflation Factor to ensure the independence of the predictors. (3) For each midface variable, the data was fit to a multiple linear regression model. All variables were incorporated into the multiple linear regression model using a stepwise method, where variables with high significance were systematically eliminated. (4) To verify the adequacy of the models, the scale–location plot, residuals vs. fitted plot, and Q-Q plot were examined. (5) Finally, cross-validation was performed using the leave-one-out (LOO) and 3-fold methods to assess the reliability of the model.

## 3. Results

### 3.1. Morphological Characteristics of Craniofacial Region Between SC and Controls

#### 3.1.1. Demographic Data

This study included a total of 34 computed tomographic (CT) scans. The patient cohort consisted of 17 patients with SC (6 Apert syndromes, 6 Crouzon syndromes, and 5 Pfeiffer syndromes). There were 6 males and 11 females. A total of 17 age-matched non-syndromic children CT scans were selected, with 7 males and 10 females. Both the SC group and control group were selected among children aged from 5 months to 11 years. The median age of the SC group was 17 months (IQR:10.00–45.5). On the other hand, the median age of the control group was 19 months (IQR:11.00–49.25). There was no statistical difference in age between the two groups (*p* = 0.652).

#### 3.1.2. Craniofacial Morphology

The overall cranial base length was shortened by 9% compared to the control group, primarily due to a significant reduction in the middle cranial fossa (S-SO, 73%, *p* < 0.01). The anterior cranial base was slightly shorter but did not reach statistical significance. The N-S-SO angle was sharper, indicating that the sphenoid body rotated (SO points moved downwards). Additionally, the N-SO-BA angle decreased, suggesting a trend towards kyphosis in the cranial base. In the midface, the length (ANS-PNS) and zygomatic bone length (ZMs-ZTi) were significantly reduced by 15% and 11%, respectively, compared to the control group. However, the midface width (ZFL-ZFR) increased by 10%. The midface height (N-ANS), maxillary width (ZMsL-ZMsR), and zygomatic height (ZX-ZF) were close to normal values. The results of the craniofacial morphology are shown in Figure 1 and Table 2.

### 3.2. The Predictive Tool of Skeletal Growth

#### 3.2.1. Variables Selection

In this study, cranial base variables with no statistically significant difference compared to the control group were selected as the predictor variables. As shown in Table 2, four variables were chosen: S-N, SO-BA, N-S-BA and S-SO-BA. The age factor was also added as a predictor variable. Three midfacial variables with significant statistical difference were selected as response variables namely: ANS-PNS, ZFL-ZFR and ZMs-ZTi. All selected variables were commonly used markers in the multiple literatures and were easily identifiable and labeled in practice.

#### 3.2.2. Correlation Analysis and Multicollinearity Checking

The results of the correlation analysis between the predictor and response variables are shown by a correlation plot in Table 3. S-N, SO-BA, and age and response variables exhibited a strong positive correlation. There were moderate to weak positive correlations between N-S-BA and response variables and S-SO-BA and response variables.

The multicollinearity check was performed on all selected cranial base and age variables to assess their independence in the regression model. The Variance Inflation Factor (VIF) scores for S-N, SO-BA, N-S-BA, S-SO-BA, and age were 4.683, 3.626, 1.520, 1.282, and 5.890, respectively. These results suggested that the variables exhibited acceptable levels of multicollinearity. The scores for the variables are presented in Figure 2.

#### 3.2.3. Model Fitting and Adequacy

The scale–location plot was used to assess the homoscedasticity of the residuals. For the midface length (ANS-PNS) model, there was a relatively even distribution of the square root of standardized residuals around the fitted values, with no clear pattern or funnel shape. The model of the midface width (ZFL-ZFR) and zygoma length (ZMs-ZTi) showed similar patterns, which suggested homoscedasticity. The results are shown in Figure 3. The Residuals vs. fitted plot was used to check the linearity. The results for all three models show no clear patterns, with residuals scattered randomly around the horizontal line, supporting the linearity assumption (Figure 4). The Q-Q plot was used to assess the normality of the residuals. For all three models, the residuals were approximately normally distributed, with only minor deviations at the tails (Figure 5). The regression models for ANS-PNS, ZFL-ZFR, and ZMs-ZTi were appropriately specified, and the key assumptions of regression analysis were reasonably met for these variables.

#### 3.2.4. Multiple Regression Models and Cross-Validation

All variables were incorporated into the multiple linear regression model using a stepwise method, where variables with high significance were systematically eliminated. Three predictive models were deemed reliable for the midfacial region: (1) ANS-PNS = 23.976 + 0.139 S-N + 0.545 SO-BA − 0.120 N-S-BA + 0.078 S-SO-BA + 0.051 age; (2) ZFL-ZFR = −15.618 + 0.666 S-N + 0.241 N-S-BA + 0.155 S-SO-BA + 0.121 age; and (3) ZMs-ZTi = −14.403 + 0.765 SO-BA + 0.266 N-S-BA + 0.111 age. These models have an R2 of 0.978, 0.903, and 0.878, respectively (Table 4). This indicates that the percentage of variance for the midface explained by the cranial base variables in the regression model is high, ranging from 87.8% to 97.8% of the total variance. The adjusted R2 was also reported to signify the significance of the predictors included in the model. The cross-validation results further confirmed the model’s robustness, with low root mean square error (RMSE) and mean absolute error (MAE) values, particularly for the midface length.

## 4. Discussion

In this study, we compared the cranial base and midfacial morphology in patients with SC to control subjects. Through the use of stable cranial base measurements, this study predicted changeable midfacial measurements, including the midfacial length and width, as well as the zygomatic bone length using a multiple linear regression model, which serves as a potential foundation for clinicians to enhance the precision of midfacial surgical corrections, thereby assisting in surgical planning and individualized treatment strategies, which could potentially improve surgical precision and outcomes.

The cranial base, as a crucial platform for brain development, contains multiple growth centers that drive the growth of both the skull and face [11]. The synostotic process of the cranial base influences the growth of the facial complex in patients with SC [12]. The results of the current study support the conclusion. The cranial base in SC patients is notably shorter, has a downwards rotation, and is more kyphotic, which is predominantly attributed to a reduction in the middle and posterior cranial fossa. This observation aligns with findings from previous studies by Cha et al. [13] and Lu et al [8,14]. Due to the premature fusion of the cranial base cartilage, such as at the S-SO (sella-occipital) and S-ES (sella-ethmoid) synchondroses, the development of the cranial base in the sagittal direction is restricted. The growing brain then forces compensatory growth at the remaining open sutures and synchondroses, which affects the angles of the cranial base. The downward rotation and kyphosis of the cranial base, as a compensatory mechanism for the shortened cranial base, allow for an increased volume to accommodate a larger brain. The abnormal cranial base morphology significantly impacts midfacial development. The shortening of the middle and posterior cranial fossae leads to evident same-direction reductions in the maxillary length (ANS-PNS) and zygomatic length (ZMs-ZTi), contributing to midfacial hypoplasia. The observed increase in the midfacial width (ZFL-ZFR) may also represent a compensatory growth mechanism. These findings are consistent with previous work that observed similar midfacial morphology in patients with SC [5,15].

The cranial base forms primarily through endochondral ossification, with contributions from mesodermal and neural crest cells, involving growth centers and sutures that remain active until late adolescence. These growth centers drive the development of the whole skull and face. The paper by Flaherty et al. underscores the crucial role that cranial base angles and lengths play in craniofacial morphology. Altered cranial base angles and lengths can lead to an underdeveloped and incorrectly positioned midface. These changes disrupt the normal growth trajectory of the midface, leading to functional and aesthetic issues such as dental arch alignment, malocclusion, and airway obstruction which are often observed in various craniosynostosis disorders [16].

Chat et al. provided detailed observations on an Apert syndrome patient, illustrating how a shorter anterior cranial base impacts midfacial development such as dental arch alignment, occlusion, and airway obstruction, which further complicates the clinical management of these patients [13]. The linear growth of part of the anterior cranial base, as discussed in Stramrud’s study, is crucial for maintaining proper spatial relationships between cranial and facial structures during growth [17]. This supports the hypothesis that the cranial base can serve as a linear regression predictor for midface development. In the current study, the S-N (sella-nasion) demonstrated strong correlations with the midface sagittal length, transverse width, and zygomatic bone length. The SO-BA is a sloping bony structure located at the posterior cranial base and is associated with the anterior position of the maxilla [18]. This study’s morphometric analysis showed that this part of the posterior cranial base was connected through the temporal bone, which affects the development and placement of cheekbones. This indirectly affects the development of the maxilla. As demonstrated in this current study, there was a strong correlation with all midfacial measurements.

The deviation in the cranial base angle indicates disruptions in the coordinated growth of the anterior and posterior cranial base, which is essential for normal facial development. Changes in the cranial base angle are correlated with the spatial relationship between the cranial base and the facial skeleton [19,20]. These angles can serve as a useful diagnostic marker for identifying and assessing the severity of structural discrepancy in SC [21]. From the results of the current study, the cranial base angles showed moderate to weak correlations, which are consistent with previous findings. The positive correlation between age and midfacial dimensions suggests that discrepancies in midfacial growth become more progressive with age [4]. This finding is consistent with the research of Reitsma et al., who emphasized that the retrusion of the midface increases with age [4]. It is worth mentioning that although the degree of craniofacial deformity increased over time, the pattern was different due to the degree of fusion maturity and the timing of the craniobasal synchondrosis. This disparity can be attributed to the patent synchondroses in Apert infants, which gradually undergo progressive synostosis with age, whereas in Crouzon syndrome patients, the fusion of the cranial base synchondroses occurs earlier than Apert syndrome and persists as a consistent deformity [22].

Kushida et al. demonstrated the potential of using craniofacial morphometric parameters to predict the severity of obstructive sleep apnea, which only include two craniofacial factors (maxillary intermolar distance and mandibular intermolar distance) [23]. However, the Kushida morphometric model might not be a suitable post-operative predictor in maxillomandibular surgery due to the limited application of only two craniofacial region parameters [24].

As the degree of midface hypoplasia in SC varies significantly, multiple morphometric parameters should be used and further analyzed. The craniofacial complexity in SC is caused by genetic disorders that disrupt normal osteogenic signals resulting in the premature closure of sutures, impeding anterior–posterior growth, and increasing the transverse width of multiple craniofacial regions of the midface, including the maxilla and zygomatic bones [25]. Even though Hariri et al. have provided a foundation for predicting the maxillary width using the whole cranial base, the study had limited age-matching between the SC patients and the control group [6]. In the current study, this limitation was addressed by ensuring age-matching and including age as a predictor in the model. Three predictive mathematical models for skeletal growth in the midface region using stable cranial base variables were developed as follows: (1) midface length (ANS-PNS) = 23.976 + 0.139 S-N + 0.545 SO-BA − 0.120 N-S-BA + 0.078 S-SO-BA + 0.051 age; (2) midfacial width (ZF-ZFR) = −15.618 + 0.666 S-N + 0.241 N-S-BA + 0.155 S-SO-BA + 0.121 age; and (3) zygoma length (ZMs-ZTi) = −14.403 + 0.765 SO-BA + 0.266 N-S-BA + 0.111 age. All of these demonstrated strong performance and robustness and were validated through rigorous cross-validation techniques.

The analysis demonstrated that the cranial base angle N-S-BA shows mixed effects: negative for the midface length but positive for the midfacial width and zygoma length. In addition, the SO-BA is a strong predictor for midfacial and zygomatic length models while the anterior cranial base length (S-N) is a strong predictor for the midfacial width. These models contribute to the understanding of the relationship between cranial base development and midfacial growth, offer insights for further research and refinement of predictive models, guide tailored surgical intervention planning, and improve the precision of surgical outcomes.

The limitations of this study stem from the stringent inclusion criteria. All SC patients included had not undergone any surgical interventions to ensure that the craniofacial growth patterns and morphological assessments were not influenced by any prior surgeries. Furthermore, due to concerns about radiation exposure, CT imaging was not typically employed for young children, leading to a small sample size for both groups in this study. However, the patient and control groups were matched for age, gender, and three common subtypes (Apert, Crouzon, and Pfeiffer) which helped to ensure acceptable reliability of the study results. Future research should focus on validating these predictive models in larger cohorts. Although the current models require further external validation in larger cohorts, the proposed predictive framework holds promise for clinical translation. These regression-based models may be integrated into digital craniofacial assessment software or surgical planning platforms to provide individualized estimations of midfacial growth patterns based on stable cranial base parameters. Such predictive tools could assist clinicians in determining optimal surgical timing and scope, ultimately contributing to more precise and patient-specific craniofacial interventions in syndromic craniosynostosis management.

## 5. Conclusions

This study demonstrated that patients with SC have a shortened, rotated, and kyphotic cranial base, a decreased midfacial and zygomatic length, as well as an increased midfacial width. This study proposed three predictive models of the midfacial region based on stable cranial base variables: (1) midface length (ANS-PNS) = 23.976 + 0.139 S-N + 0.545 SO-BA − 0.120 N-S-BA + 0.078 S-SO-BA + 0.051 age; (2) midfacial width (ZF-ZFR) = −15.618 + 0.666 S-N + 0.241 N-S-BA + 0.155 S-SO-BA + 0.121 age; and (3) zygoma length (ZMs-ZTi) = −14.403 + 0.765 SO-BA + 0.266 N-S-BA + 0.111 age. The study findings emphasize the need for comprehensive cranial base assessments in clinical practice to ensure accurate predictions and better management of patients with SC.

## Figures and Tables

**Figure 1 diagnostics-15-01640-f001:**
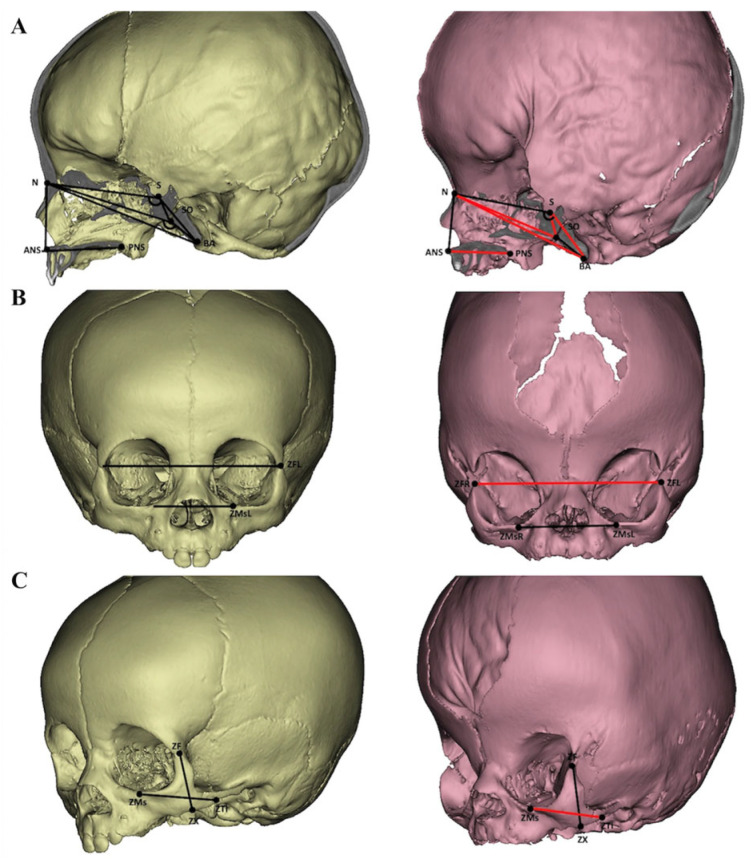
Females aged 10 months were used in both groups (left—control, right—Apert syndrome). The red lines and angles illustrated the changed variables compared to the control. Sagittal view (**A**), frontal view (**B**), and lateral view (**C**).

**Figure 2 diagnostics-15-01640-f002:**
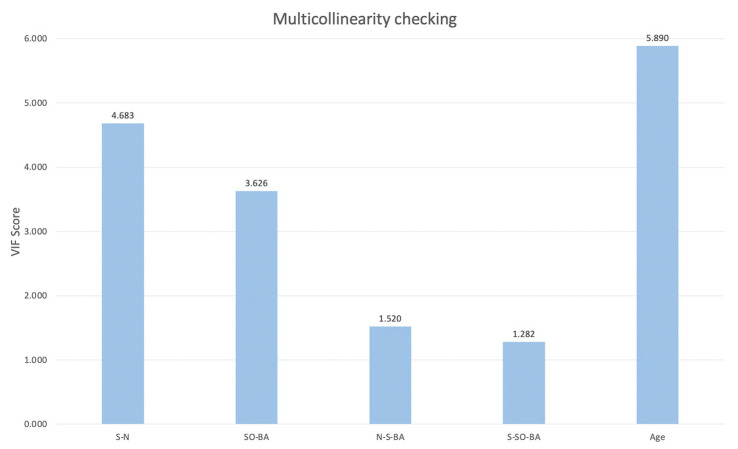
The results of the VIF score for selected response variables.

**Figure 3 diagnostics-15-01640-f003:**
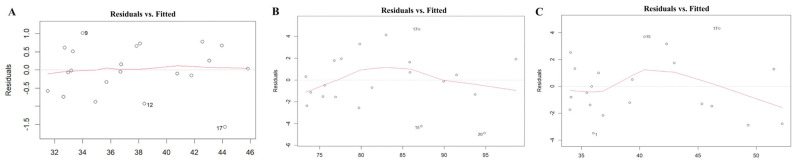
The homoscedasticity of ANS-PNS (**A**), ZFL-ZFR (**B**), and ZMs-ZTi (**C**).

**Figure 4 diagnostics-15-01640-f004:**
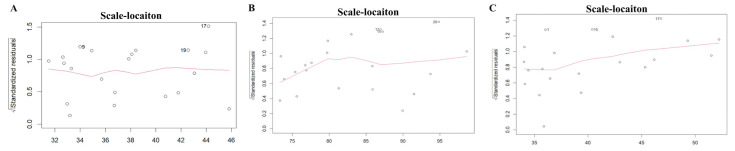
The linearity of the relationship of ANS-PNS (**A**), ZFL-ZFR (**B**), and ZMs-ZTi (**C**).

**Figure 5 diagnostics-15-01640-f005:**
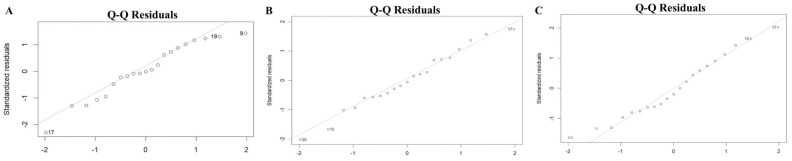
The normality of the residuals of ANS-PNS (**A**), ZFL-ZFR (**B**), and ZMs-ZTi (**C**).

**Table 1 diagnostics-15-01640-t001:** The definition of the cranial base and midface landmarks.

Abbreviations	Name	Definition
ANS	Anterior nasal spine	It is the anterior tip of the sharp bony process of the maxilla
BA	Basion	The most inferior and posterior point on the most anterior margin of the foramen magnum
N	Nasion	The most anterior and canter point where the frontal and two nasal bones meet
PNS	Posterior nasal spine	The most posterior midpoint of the posterior nasal spine of the palatine bone
S	Sella	The lowest and center point of the hypophyseal fossa
SO	Spheno-occipital synchondrosis	The most anterior point on the midline of the occipital bone at the spheno-occipital synchondrosis
ZF(L/R)	Zygomaticofrontal suture	The intersection points of the zygomatic frontal suture on both sides
ZMs(L/R)	Zygomaticomaxillary suture	The point at the zygomaticomaxillary suture on the inferior orbital rim on each side
ZTi(L/R)	Zygomaticotemporal suture	The inferior point of the zygomaticotemporal suture
ZX(L/R)	Zygomatic point	The most middle point of the inferior border of the zygomatic bone

**Table 2 diagnostics-15-01640-t002:** The summary statistics of the craniofacial morphology in the SC and control group. Their absolute difference (D), percentage of difference, and the *p*-value of the Mann–Whitney U test are also presented.

	SC		Control				
Variables	Median	IQR	Median	IQR	D	D (%)	*p*-Value
Cranial base							
N-BA	67.22	(63.41, 73.25)	73.65	(69.71, 82.89)	−6.43	91%	0.014 *
N-S-BA	138.21	(128.00, 141.11)	140.80	(137.66, 144.73)	−2.58	98%	0.054
N-S-SO	117.48	(108.09, 127.74)	127.09	(120.63, 130.54)	−9.60	92%	0.031 *
N-SO	52.90	(50.48, 61.64)	59.87	(56.29, 64.27)	−6.97	88%	0.013 *
N-SO-BA	157.75	(151.64, 162.45)	166.16	(162.17, 169.76)	−8.41	95%	0.002 **
S-BA	23.83	(21.77, 26.12)	27.82	(25.73, 31.39)	−3.99	86%	0.001 **
S-N	47.10	(44.72, 54.60)	49.86	(47.06, 55.64)	−2.76	94%	0.264
S-SO	10.53	(9.65, 11.20)	14.34	(12.85, 15.33)	−3.81	73%	0.000 **
S-SO-BA	146.52	(143.66, 156.12)	151.57	(147.45, 153.66)	−5.05	97%	0.195
SO-BA	14.56	(12.59, 16.51)	14.28	(13.09, 17.35)	0.29	102%	0.601
Midface							
ANS-PNS	29.74	(27.79, 34.39)	35.19	(33.20, 38.50)	−5.45	85%	0.003 **
N-ANS	32.29	(27.34, 38.50)	32.08	(29.04, 38.86)	0.21	101%	0.471
ZMsL-ZMsR	42.61	(38.94, 48.54)	42.43	(38.65, 47.28)	0.18	100%	0.986
ZFL-ZFR	87.11	(80.76, 94.12)	79.05	(74.19, 87.00)	8.06	110%	0.034 *
ZMs-ZTi	32.11	(30.01, 40.08)	36.23	(34.44, 43.01)	−4.12	89%	0.044 *
ZX-ZF	31.11	(28.84, 33.70)	30.80	(27.47, 34.51)	0.31	101%	0.857

IQR: interquartile range (25–75%); D(mm): difference—SC median–control median; %: percentage—SC median/control median; *p*: Mann–Whitney U test, * *p* < 0.05, ** *p* < 0.01.

**Table 3 diagnostics-15-01640-t003:** The correlation coefficients of selected midface and cranial base variables.

Variables	S-N	SO-BA	N-S-BA	S-SO-BA	Age
ANS-PNS	0.891	0.916	−0.605	−0.315	0.951
ZFL-ZFR	0.884	0.816	−0.384	−0.244	0.912
ZMs-ZTi	0.854	0.856	−0.329	−0.233	0.902

**Table 4 diagnostics-15-01640-t004:** The predictive models with their respective R^2^ values as a measure of the model’s goodness of fit. The performance of these models was evaluated using cross-validation (CV) techniques, namely the leave-one-out (LOO) and 3-fold methods. The reliability of these models was quantified by the root mean square error (RMSE) and mean absolute error (MAE).

Measurement	Model	R^2^	Adjusted R^2^	LOOCV			3-Fold		
				RMSE	R^2^	MAE	RMSE	R^2^	MAE
Midface Length	**ANS-PNS** = 23.976 + 0.139 S-N + 0.545 SO-BA − 0.120 N-S-BA + 0.078 S-SO-BA + 0.051 Age	0.978	0.971	0.904	0.959	0.736	1.058	0.945	0.843
Midface Width	**ZFL-ZFR** = −15.618 + 0.666 S-N + 0.241 N-S-BA + 0.155 S-SO-BA + 0.121 Age	0.903	0.879	3.191	0.874	2.679	3.158	0.840	2.584
Zygoma Length	**ZMs-ZTi** = −14.403 + 0.765 SO-BA + 0.266 N-S-BA + 0.111 Age	0.878	0.857	2.824	0.798	2.409	3.720	0.947	3.076

## Data Availability

All data can be accessed and are available upon request via email to the corresponding author.

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
