# Peer review of "Enhancing Predictive Tools for Skeletal Growth and Craniofacial Morphology in Syndromic Craniosynostosis: A Focus on Cranial Base Variables"

_diagnostics, 2025, doi:10.3390/diagnostics15131640_

Round 1

Reviewer 1 Report

Comments and Suggestions for Authors

Dear Authors,
I read with interest the research that proposes a tool that uses the characteristics of the skull base in syndromic patients with craniosynostosis to predict midfacial growth.
I ask you to clarify some points that I did not understand well:
- Did you calculate the sample size to evaluate the number of patients to enroll in the study?
- The total patients are 34 (line 121), but in the description of the method you state that you examined the CTs of 17 syndromic patients and 17 normal patients of the same age, and 21 normal children under 12 years old. I did not understand how these last 21 patients were used.
- Furthermore, the experimental group and the control group, which have a similar age, are less than two years old. This is understandable because the syndromic patients must not have been operated on yet. Why is the age limit for enrollment 11 years? How many patients are older than 2 years old? Could they have different characteristics, since the growth of the maxilla occurs during childhood?
-Are there differences between patients with Apert, Crouzon and Pfeiffer?
-I understand the difficulty of finding an adequate sample for the study, but if the data are not sufficient to demonstrate the validity of the method, this should be clarified in the text. Perhaps we could talk about a pilot study.
-Why did the healthy children have CT? I think it should be clarified to avoid ethical doubts.
- I think the characteristics of the healthy children used as controls should be better specified: did they have harmonious growth of the upper jaw and mandible and a good dental occlusion?
- In the discussion, at line 212, you write "thereby reducing the need for multiple surgeries": this does not seem to me to be an evidence-based sentence.

Author Response

Comment 1: Did you calculate the sample size to evaluate the number of patients to enroll in the study?

Response 1: Yes, Sample size calculation was performed using G*Power 3.1 (Faul et al., 2007; Kang, 2021) determine the appropriate number of patients to enroll in our study. We set the effect size, f² =1.0 due to high observed R² value, significance level at α = 0.05 with a desired power of 0.90 and 5 predictors.  The calculated minimum sample size was 23. However, we are able to increase the control samples from 17 to 21 only, due to the limited sample from our retrospective archive. While slightly below the ideal target, the size of control sample still provides adequate statistical power at approximately 0.8. We have added this information in the methodology (lines 90-102) section.

Comment 2: The total patients are 34 (line 121), but in the description of the method you state that you examined the CTs of 17 syndromic patients and 17 normal patients of the same age, and 21 normal children under 12 years old. I did not understand how these last 21 patients were used.

Response 2: We apologize for the confusion caused. The 34 patients mentioned include the 17 SC patients and 17 age-matched controls for morphology analysis. We then increase the control samples from 17 to 21. These 21-control sample were solely used for model development rather than for group comparison in the morphological analysis. We have now clarified this point in the revised Materials and Methods section (Lines 90-102).

Comment 3:Furthermore, the experimental group and the control group, which have a similar age, are less than two years old. This is understandable because the syndromic patients must not have been operated on yet. Why is the age limit for enrollment 11 years? How many patients are older than 2 years old? Could they have different characteristics, since the growth of the maxilla occurs during childhood?

Response 3: Thanks for the insightful comment. We chose an upper age limit of 12 years because (1) cranial base synchondroses such as the Spheno-occipital synchondrosis and Spheno-ethmoidal synchondrosis can remain patent into late childhood, thus allowing continued growth and capturing a broader spectrum of cranial base; (2) the oldest patient available in our dataset was 12 years old, so the cut-off of 12 was selected as a natural upper boundary of children; (3) we think that the inclusion of children across a wider age range enhances variability and heterogeneity in the dataset, which is beneficial for constructing generalizable predictive models. In our study population, 12 of the 17 SC patients were younger than 2 years old, while 5 patients were older than 2 years but younger than 12 years. Similarly, in the control group, 10 children were younger than 2 years, and 7 were between 2 and 12 years old. While craniofacial growth dynamics differ between infants and older children, we accounted for age as an independent covariate in the regression models to adjust for growth-related differences. This approach allowed us to partially control for age-dependent effects in craniofacial morphology.

Comment 4:Are there differences between patients with Apert, Crouzon and Pfeiffer?

Response 4: Yes, variations exist among different SC subtypes. However, the primary objective of this study was not to compare individual syndromic subtypes, but rather to establish generalizable predictive models for midfacial growth based on cranial base variables across SC patients as a group. Given the limited sample size, subgroup analyses by syndrome type would lack sufficient statistical power and risk overfitting.

Comment 5: I understand the difficulty of finding an adequate sample for the study, but if the data are not sufficient to demonstrate the validity of the method, this should be clarified in the text. Perhaps we could talk about a pilot study.

Response 5: We greatly appreciate the reviewer’s understanding regarding the challenges of patient recruitment in this field. Although our sample size is slightly below the ideal target, the 21 control samples still provide adequate statistical power at approximately 0.8. To further strengthen the analysis, thorough procedure of multiple linear regression modeling was carried out, incorporating leave-one-out and 3-fold cross validation. The models consistently demonstrated low RMSE values, indicating good predictive performance despite the limited sample size. We have addressed in this Discussion section, line 395-399.

Comment 6: Why did the healthy children have CT? I think it should be clarified to avoid ethical doubts.

Response 6: We sincerely thank the reviewer for raising this important ethical concern. The CT scans of the healthy control group were retrospectively collected from hospital imaging archives. All control subjects were non-syndromic children who underwent cranial CT for non-craniofacial clinical indications (such as headache or neurological evaluation). No healthy subject underwent CT solely for research purposes. And we have replaced healthy with non-syndromic.

Comment 7: I think the characteristics of the healthy children used as controls should be better specified: did they have harmonious growth of the upper jaw and mandible and a good dental occlusion?

Response 7: Yes, the non-syndromic children selected as controls had normal craniofacial structures without any craniofacial deformities, midfacial hypoplasia, or dental malocclusion, based on clinical records and radiological assessment by experienced maxillofacial surgeons and radiologists. All control subjects were non-syndromic children who underwent cranial CT for non-craniofacial clinical indications (such as headache or neurological evaluation). And we have replaced healthy with non-syndromic.

Comment 8: In the discussion, at line 212, you write "thereby reducing the need for multiple surgeries": this does not seem to me to be an evidence-based sentence.

Response 8: We appreciate this valuable observation. We have revised the sentence to read: “These models may assist in surgical planning and individualized treatment strategies, which could potentially improve surgical precision and outcomes." This revision has been made in the Discussion section (Line 294-295) to ensure it reflects the current level of evidence without overstatement.

Reviewer 2 Report

Comments and Suggestions for Authors

This study aims to develop a predictive tools for midfacial growth using stable cranial base metrics, a novel and practical contribution to the pediatric maxillofacial surgery.

With high methodological rigor, demonstrating strong predictive performance on a real-world issues: midface hypoplasia, surgical planning challenges, and outcome variability in SC patients.

This predictive model could be a feasible aid in surgical timing and scope decisions.

I suggest to improve clarity and fluency with a professional English language edit and moreover add in the discussion more on developmental biology behind these specific base variables selections.

Futhermore I'd suggest to discuss how these models might be translated into tools clinicians can use.

Author Response

Comment 1: I suggest improving clarity and fluency with a professional English language edit and moreover add in the discussion more on developmental biology behind these specific base variables selections.

Response 1: Thank you for this helpful suggestion.The selection of the cranial base variables (S-N, SO-BA, N-S-BA, and S-SO-BA) was based on a combination of both statistical screening and developmental biology considerations: (1) as described in the Materials and Methods (Line 109-110), these cranial base landmarks (N, S, SO, BA) are frequently employed in syndromic craniosynostosis research due to their reproducibility and anatomical stability across different stages of development, even under pathological conditions; (2) these four variables were identified as statistically stable predictors after comparative morphometric analysis between the syndromic patients and age-matched controls; specifically, they exhibited no statistically significant differences between the two groups (Table 2), indicating that these cranial base parameters remain relatively preserved across pathological and normal craniofacial growth; (3) the selected parameters also reflect important underlying developmental biology: The cranial base forms primarily through endochondral ossification, with contributions from mesodermal and neural crest cells, involving growth centers and sutures that remain active until late adolescence. These growth impacts midfacial development and position, which has already been discussed in detail in the Discussion section (Lines 296-340). Altered cranial base angles and lengths can lead to an underdeveloped and incorrectly positioned midface, which has already been discussed in detail in the Discussion section (Lines 341-356).

Comment 2: Furthermore I'd suggest discussing how these models might be translated into tools clinicians can use.

Response 2: Thank the reviewer for this important translational suggestion. In response, we have added a new paragraph to the end of the Discussion section (Lines 399–407), which reads as follows: ”Although the current models require further external validation in larger cohorts, the proposed predictive framework holds promise for clinical translation based on the matched sample used in this study. These regression-based models may be integrated into digital craniofacial assessment software or surgical planning platforms to provide individualized estimations of midfacial growth patterns based on stable cranial base parameters. Such predictive tools could assist clinicians in determining optimal surgical timing and scope, ultimately contributing to more precise and patient-specific craniofacial interventions in syndromic craniosynostosis management.”

Reviewer 3 Report

Comments and Suggestions for Authors

The article under review is devoted to the interesting problem of predicting the course of complex congenital bone malformations in growing children. More specifically, the authors use an original method of craniometric ratios, which they proposed earlier, and in this study applied it to a larger group of children with syndromic craniosynostosis. The research showed the presence of a correlation between the craniometric parameters and age. Despite the obvious advantages (originality of the calculation method, statistically confirmed reliability), there are several comments from the reviewer that should be taken into account before publication.

  1. Craniosynostosis is a common condition, including a part of genetic syndromes. The authors generalize their findings to craniosynostoses, although in fact they only evaluate patients with Apert, Crouzon and Pfeiffer syndromes. It seems important to note this, if not in the title, then at least in the purpose and conclusion of the article.
  2. In continuation, the authors did not indicate whether a genetic study was conducted to confirm the diagnosis and what its results were. The introduction to the article mentions that mutations in the FGFR gene are the cause of craniosynostosis. In this case, it is necessary to clarify which specific genes are in question (usually FGFR 1 and 2), since fibroblast growth factor receptors are involved in a huge number of processes and pathogenic variants in the genes of these proteins are responsible for a large number of diseases in addition to craniosynostoses.
  3. It is necessary to clarify the source and the reasons for performing tomography on patients in the control group (ethical aspects)
  4. It is advisable to provide specific examples of the possible use of the obtained data
  5. In the figure 1A demonstrating the positions of craniometric points in the patient, the points on the facial part are not clearly indicated (gray stripe in the figure)

The article is recommended for publication after appropriate corrections.

Author Response

Comment 1: Craniosynostosis is a common condition, including a part of genetic syndromes. The authors generalize their findings to craniosynostoses, although in fact they only evaluate patients with Apert, Crouzon and Pfeiffer syndromes. It seems important to note this, if not in the title, then at least in the purpose and conclusion of the article.

Response 1: We sincerely thank the reviewer for this valuable observation. These three subtypes are the most common forms of SC, accounting for the majority of syndromic cases, which constitute the predominant syndromic forms encountered in clinical practice and represent the core population for developing predictive models. We have made the following revisions in the purpose and conclusion parts: 1) lines 73-74, “craniosynostosis who specifically diagnosed with Apert, Crouzon, and Pfeiffer syndromes.” 2) lines 419-420, “patients with these common subtypes of syndromic craniosynostosis (Apert, Crouzon and Pfeiffer syndromes).”

Comment 2: In continuation, the authors did not indicate whether a genetic study was conducted to confirm the diagnosis and what its results were. The introduction to the article mentions that mutations in the FGFR gene are the cause of craniosynostosis. In this case, it is necessary to clarify which specific genes are in question (usually FGFR 1 and 2), since fibroblast growth factor receptors are involved in a huge number of processes and pathogenic variants in the genes of these proteins are responsible for a large number of diseases in addition to craniosynostoses.

Response 2: we have revised the Introduction and Materials and Methods to specify:"Mutations in FGFR2 are primarily implicated in Apert and Crouzon syndromes, while both FGFR1 and FGFR2 mutations may contribute to Pfeiffer syndrome” in lines 48-50. “The diagnoses of Apert, Crouzon, and Pfeiffer syndromes were made clinically by experienced craniofacial surgeons based on characteristic phenotypic features and CT imaging. Genetic confirmation was not systematically available for all cases. However, these syndromes are typically associated with mutations in FGFR2 (Apert and Crouzon) and FGFR1/FGFR2 (Pfeiffer), as noted in the Introduction” in lines 83-87.

Comment 3: It is necessary to clarify the source and the reasons for performing tomography on patients in the control group (ethical aspects).

Response 3: Sorry for the ambiguity caused by the incorrect expression. we have added a clarification regarding the control group: the non-syndromic children selected as controls had normal craniofacial structures without any craniofacial deformities, midfacial hypoplasia, or dental malocclusion, based on clinical records and radiological assessment by experienced maxillofacial surgeons and radiologists. All control subjects were non-syndromic children who underwent cranial CT (such as headache, seizures or neurological evaluation for unrelated complaints). And we have replaced healthy with non-syndromic.

Comment 4: It is advisable to provide specific examples of the possible use of the obtained data.

Response 4: We sincerely thank the reviewer for highlighting this valuable translational aspect. In the Discussion (last paragraph), we have expanded the clinical application as follows: “Although the current models require further external validation in larger cohorts, the proposed predictive framework holds promise for clinical translation based on the matched sample used in this study. These regression-based models may be integrated into digital craniofacial assessment software or surgical planning platforms to provide individualized estimations of midfacial growth patterns based on stable cranial base parameters. Such predictive tools could assist clinicians in determining optimal surgical timing and scope, ultimately contributing to more precise and patient-specific craniofacial interventions in syndromic craniosynostosis management.”

Comment 5: In the figure 1A demonstrating the positions of craniometric points in the patient, the points on the facial part are not clearly indicated (gray stripe in the figure).

Response 5: We have now updated Figure 1A to improve the visibility of facial craniometric landmarks.

Round 2

Reviewer 1 Report

Comments and Suggestions for Authors

Dear Autuors, I am satisfied from the changes in the manuscript. Now it is clear and ready for publication

Reviewer 2 Report

Comments and Suggestions for Authors

the manuscript can be accepted in the present form.